# Between and Within-Country Variations in Infant and Young Child Feeding Practices in South Asia

**DOI:** 10.3390/ijerph19074350

**Published:** 2022-04-05

**Authors:** Md. Tariqujjaman, Md. Mehedi Hasan, Mustafa Mahfuz, Tahmeed Ahmed, Muttaquina Hossain

**Affiliations:** 1Nutrition and Clinical Services Division, icddr,b, Dhaka 1212, Bangladesh; mustafa@icddrb.org (M.M.); tahmeed@icddrb.org (T.A.); muttaquina@icddrb.org (M.H.); 2Institute for Social Science Research, The University of Queensland, Indooroopilly, QLD 4068, Australia; m.m.hasan@uq.net.au; 3Australian Research Council Centre of Excellence for Children and Families over the Life Course, The University of Queensland, Indooroopilly, QLD 4068, Australia

**Keywords:** infant and young child feeding practices, South Asia, variations, sociodemographic subgroups

## Abstract

This study aimed to explore variations in Infant and Young Child Feeding (IYCF) practices between different South Asian Countries (SACs) and within their sociodemographic characteristics including place of residence, mother age, mother education, child sex, and wealth quintiles within the SACs. We extracted 0–23 months age children’s data from the nationally representative survey of Afghanistan, Bangladesh, India, Maldives, Nepal, and Pakistan. Among all SACs, the early initiation of breastfeeding (EIBF) practice was 45.4% with the highest prevalence in the Maldives (68.2%) and the lowest prevalence in Pakistan (20.8%). Exclusive breastfeeding (EBF) practice was 53.9% with the highest prevalence in Nepal (67%) and the lowest prevalence in Afghanistan (42%). Only 13% of children had a minimum acceptable diet (MAD), with the highest prevalence in the Maldives (52%) and the lowest prevalence in India (11%). We found higher IYCF practices among the mothers with secondary or higher levels of education (EIBF: 47.0% vs. 43.6%; EBF: 55.5% vs. 52.0%; MAD: 15.3% vs. 10.0%), urban mothers (MAD: 15.6% vs. 11.8%), and mothers from the richest households (MAD: 17.6% vs. 8.6%) compared to the mothers with no formal education or below secondary level education, rural mothers and mothers from the poorest households, respectively. Mothers from the poorest households had better EIBF, EBF, and continued breastfeeding at 1-year (CBF) practices compared to the mothers from the richest households (EIBF: 44.2% vs. 40.7%; EBF: 54.8% vs. 53.0%; CBF: 86.3% vs. 77.8%). Poor IYCF practices were most prevalent in Afghanistan, Pakistan, and India.

## 1. Introduction

The first two years of life are considered to be the “window of opportunity” for a child’s physical, social, emotional, cognitive, and brain development [1]. Appropriate Infant and Young Child Feeding (IYCF) practices are imperative for a child’s physical and cognitive development [2]. These include timely initiation of breastfeeding, exclusive breastfeeding for the first six months of life, the continuation of breastfeeding for two years, and the introduction of nutritionally adequate, safe, and age-appropriate solid, semisolid, or soft foods at six months of age [3].

South Asia is one of the hubs for poor nutritional status and suboptimal and inappropriate IYCF practices [4]. Nearly half of the global undernourished children are living in South Asian Countries (SACs) [5]. Besides, more than half of the children were not timely initiated to breastfeeding (61%) or exclusively breastfeeding (54%) in South Asia [6]. Only breastfeeding is not sufficient to meet all the nutritional requirements of children while growing. An additional diet is recommended to children from six months of age to maintain optimum growth and development [7]. Unfortunately, only half of the children in SACs were started to feeding complementary foods at six to eight months of their age as recommended [8]. Insufficient quantities and inadequate qualities of complementary foods and inappropriate feeding practices have a detrimental effect on a child’s health [9,10,11].

Significant disparities in terms of household poverty and parental education exist in complementary feeding practices in some countries such as Bangladesh, India, Pakistan, and Nepal [12]. However, less is known about the disparities for the full suits of WHO-core IYCF indicators in SACs individually or as a whole across other dimensions of inequality such as child’s gender, mother’s age, mother’s education, wealth quintiles, and place of residence. This clearly states the extent to which the between and within-country disparities exist in IYCF practices in SACs remain unexplored. The pooled regional and country-specific estimates and their subpopulation are particularly important for targeting locations/populations, allocating resources, designing and implementing interventions, and revisiting policies at the regional and country levels. The lack of evidence on all core IYCF indicators at regional or country levels hinders the design, implementation, and promotion of IYCF practices at a greater level in SACs. This study, therefore, aims to explore the between and within-country variations of WHO’s core IYCF indicators in SACs. The findings generated from this study would help the program managers, academia, development partners, and policymakers to set priorities and formulate appropriate strategies targeting vulnerable countries and sociodemographic vulnerable groups to improve IYCF practices.

## 2. Materials and Methods

### 2.1. Data Sources

We used nationally representative macro-level (aggregated) data from the most recent cross-sectional surveys conducted during 2015–2018 under the Demographic and Health Survey (DHS) program in Afghanistan, Bangladesh, India, Maldives, Nepal, and Pakistan. Together we called these countries SACs as per DHS classification [13]. We were unable to include other SACs such as Bhutan and Sri Lanka due to the restrictions in public use of DHS data. The DHS program provides population-based information on a wide range of monitoring and impact evaluation indicators in the areas of population, health and nutrition, and their potential determinants. The DHS generally applies a uniform procedure by using a stratified multistage cluster sampling technique to conduct the surveys. The detailed information of DHS methodology, sampling procedure, and questionnaires have been published elsewhere [14,15,16,17,18,19].

### 2.2. Study Population

The IYCF practices are generally measured for the youngest children of 0–23 months of age by asking a prescribed set of questions to their mothers. Hence, we excluded the children of more than two years of age and their mothers. Additionally, we excluded observations with any missing values in our studied variables. Therefore, the study population of this study was the youngest children under two years of age and their mothers aged 15 to 49 years old.

### 2.3. Measurement of IYCF Indicators

In this study, we examined variations in the WHO’s eight-core age-specific IYCF indicators; which included early initiation of breastfeeding (EIBF), exclusive breastfeeding under 6 months (EBF), continuing breastfeeding at 1 year (CBF), the introduction of solid, semisolid, or soft foods (ISSSF), minimum dietary diversity (MDD), minimum meal frequency (MMF), minimum acceptable diet (MAD), and consumption of iron-rich or iron-fortified foods (CIRF). All these indicators were dichotomous (yes/no). The DHS collected the information from mothers on IYCF indicators by the 24-h dietary recall questionnaire. We used the standard definitions of WHO for constructing the core IYCF indicators [3]. The detailed definition and calculation of the IYCF indicators are presented in Appendix A. In this study, the IYCF indicators were calculated taking only the complete observations for each indicator excluding the missing observations for the respective indicators.

### 2.4. Exposure Variables

We included a spectrum of sociodemographic equity dimensions including the sex of the indexed child (male and female), age and education of their mother, place of residence (rural and urban), and wealth quintiles (poorest (1st quintile), poorer, middle, richer, and richest (5th quintile)). We used the variables that the DHS constructed for defining these equity groups. The DHS constructed the household wealth quintiles based on household characteristics and ownership of assets by principal component analysis [20]. We further stratified mothers by their age as 15–24, 25–34 years, and ≥35 years, and educational status as below secondary (no education and primary) and secondary or above (secondary or higher). The mother’s education was categorized considering the distribution of the level of education (low percentage of illiterate-primary and higher merge into below secondary and secondary or higher, respectively).

### 2.5. Statistical Analyses

Data analyses included descriptions of the study population. We calculated the percentage of all WHO’s eight-core IYCF indicators for all SACs from the original survey data. Further, the percentages of IYCF indicators were calculated across a spectrum of sociodemographic equity dimensions for each of the SACs. We adjusted the effect of complex survey design including country-specific sampling weights and clusters while estimating the percentages and the respective confidence intervals.

For pooled dataset, we denormalized the sampling weight [21] and created a new population-level weight by dividing the sampling weight by the denormalized weight. We also constructed a unique cluster variable by combining country and cluster numbers. The population-level weight and unique cluster were used to calculate the pooled estimates. The population-level weight was performed to avoid the effect of countries with a large population (such as India) balancing the countries with a smaller population (such as the Maldives). The Taylor series linearization approach was used to estimate the confidence intervals (where necessary) both for pooled and country-specific analyses [22]. These approaches were frequently applied in previous studies [23,24].

We calculated the absolute differences by subtracting the estimates of one group (e.g., male) from the estimates of another group (e.g., female) to examine the extent of the variation in IYCF indicators within the SACs. We applied the adjusted Wald test to test the significance of the changes in the IYCF estimates between the equity subgroups. We calculated the relative ratio by dividing the estimates of group A (e.g., poorest, rural, mother’s below secondary education and male) by the estimates of group B (e.g., richest, urban, mother’s secondary+ education and female). The relative ratio greater than 1 denotes the IYCF practices were greater in group A compared to group B, whilst the ratio lower than 1 denotes the opposite. The chi-square test and one-way analysis of variance were performed to see any statistically significant differences of sociodemographic factors across all six countries. We performed all the statistical analyses by using statistical software Stata version 15.0 SE. We used “svy” prefix in Stata to adjust the effect of complex survey design. The *p*-value of <0.05 was considered to be statistically significant for all two-sided tests performed.

## 3. Results

### 3.1. Sociodemographic Characteristics of the Participants in SACs

The analysis included 120,830 mother–child dyads from 6 SACs. Overall, 48.7% of the mothers were aged between 25 and 34 years with the highest percentage in Pakistan (54.5%) and the lowest percentage in Bangladesh (39.3%) in this age group. More than half (52.6%) of mothers completed secondary or higher education with the highest percentage in the Maldives (84%) and the lowest percentage in Afghanistan (11%). About half of the children were female with a mean age of 11 months. One-third of the mothers lived in urban areas in all countries, except in the Maldives and Nepal where 10% and 54% of mothers respectively lived in urban areas. Overall, 25% of households belonged to the poorest quintile and 14% of households belonged to the richest quintile. Significant disparities of wealth categories across countries were observed (Table 1).

### 3.2. Pooled Prevalence of IYCF Practices in SACs

In SACs, overall, 45.4% of mothers practiced EIBF whereas the prevalence of EBF and CBF was 53.9% and 82.8%, respectively. More than half (50.2%) of the children started solid, semisolid, or soft foods at 6 to 8 months of age. Only 21.9% of children were fed diversified foods, and the prevalence of MMF was 39.9%. Only 12.8% of children had MAD and 21.4% had CIRF (Figure 1). Mothers from the poorest households had more EIBF, EBF, and CBF practices compared to the mothers from the richest households (EIBF: 44.2% vs. 40.7%; EBF: 54.8% vs. 53.0%; CBF: 86.3% vs. 77.8%). On the contrary, mothers from the poorest households had inappropriate complementary feeding (ISSSF, MDD, MMF, MAD, and CIRF) practices compared to mothers from the richest households (Figure 2A). Urban mothers had better IYCF practices (except for EBF and CBF) than rural mothers (Figure 2B). Mothers with secondary or higher education practiced better IYCF than mothers with no formal education or below secondary level education (Figure 2C).

### 3.3. Between-Country Variations of IYCF Practices in SACs

The prevalence of IYCF practices varied across the SACs. The Maldives and Nepal had the highest percentages of EIBF (68.2%) and EBF (66.5%), whereas Pakistan and Afghanistan had the lowest EIBF (20.8%) and EBF (41.9%) practice respectively. The highest prevalence of CBF was in Nepal (94.5%) and the lowest prevalence was in Pakistan (67.2%). The prevalence of ISSSF was highest in the Maldives (82%) and lowest in India (46.2%). Further, the highest prevalence of MDD, MAD, and CIRF practices was in Maldives (MDD: 71.4%, MAD: 52.1%, and CIRF: 69.5%, respectively). The lowest prevalence of MDD, MAD, and CIRF was in Pakistan (18.5%), and India (MAD: 10.7% and CIRF: 17%), respectively. In terms of MMF, the highest prevalence was in Bangladesh (79.1%) and the lowest prevalence was in India (34.6%) (Figure 1).

### 3.4. Within-Country Variations in IYCF Practices in Individual SACs across Subgroups

The IYCF practices varied across sociodemographic subgroups within SACs. We found a higher disparity in IYCF practices among the poorest vs. richest groups. The disparity was remarkable in Maldives, 44.8% for CBF (poorest: 83.9% vs. richest: 39.1%). In Pakistan, it was 18.3% for EBF (poorest: 53.9% vs. richest: 35.6%), and 26.9% for MAD (poorest: 6.3% vs. richest: 28.8%). The prevalence of breastfeeding-related indicators (EIBF, EBF, and CBF) was better among the poorest households; whereas complementary feeding indicators (ISSSF, MDD, MMF, MAD, and CIRF) were better among the richest households (Figure 2A).

The relative gaps concerning sociodemographic subgroups in IYCF practices were also visible in SACs (Figure 3A–D). The poorest/richest gaps were highest in Pakistan for EIBF and EBF; in the Maldives for CBF, MDD, MAD, and CIRF; and in Afghanistan for ISSSF and MMF (Figure 3A). The EIBF and CBF practices were better in the rural areas of Maldives and EBF in the rural areas of Bangladesh. On the contrary, the complementary feeding indicators (ISSSF, MDD, MMF, MAD, and CIRF) were better in the urban areas in Afghanistan, Bangladesh, India, and Pakistan compared to their rural counterparts (Figure 2B). The better EIBF and CBF were observed among mothers with below secondary level education in Afghanistan, Bangladesh, and the Maldives. The complementary feeding indicators were lower in below secondary educated mothers in all countries except Afghanistan for ISSSF (Figure 2C). Absolute differences in IYCF practices for male vs. female children were minimal in all SACs (Figure 2D). The relative gaps in terms of residence, mother education, and child sex in IYCF practices were presented in Figure 3B–D. The variations of IYCF practices across mothers age were presented in Appendix A.

## 4. Discussion

We explored the variations in IYCF indicators in 6 SACs—Afghanistan, Bangladesh, India, Maldives, Nepal, and Pakistan. We found suboptimal IYCF practices in EIBF, EBF, MDD, MAD, and CIRF in SACs with wider variations across the countries. The breastfeeding practices (EIBF, EBF) in Pakistan and Afghanistan were poor and India had poor complementary feeding practices (ISSSF, MMF, MAD, and CIRF) compared to other SACs. Large gaps in IYCF practices were observed in the poorest vs. richest, urban vs. rural, and less than secondary vs. secondary or higher maternal education groups.

Our pooled estimate revealed, about 45% of mothers initiated breastfeeding within one hour of childbirth. In country-specific analysis, Maldives had the highest prevalence (68.2%) of EIBF and Pakistan had the lowest (21%) which aligns with a recent finding [25,26]. The wider variation of EIBF in SACs might be due to socio-cultural, geographical location, and economic conditions among different populations [27,28,29]. We found more than half of children were exclusively breastfed in SACs. The highest EBF practice was in Nepal and the lowest was in Afghanistan. The possible reasons behind the low prevalence of EBF in SACs especially in Pakistan and Afghanistan would be the lack of political security for the proper implementation of the breastfeeding programs, poor health system policies, marketing of breast milk substitutes (BMS), and harmful traditional feeding practices [30,31]. All the SACs have laws and policies for preventing harmful practices and unethical promotion and marketing of BMS. However, most of the SACs are struggling to enforce these laws due to a lack of dedicated budget and manpower, weak monitoring system, and unpublished monitoring results [32,33]. Intensifying the market monitoring system in each SACs with proper budget and trained manpower could be a way to enforce the BMS laws and policies and could improve breastfeeding practices [34]. 

Nearly half of the infants in SACs started complementary feeding at the right age (ISSSF), which is lower than a recent LMICs study [25] and an Indian study [35]. We found only 14% of children in SACs, met the minimum acceptable diet criteria with the highest percentage was in the Maldives and the lowest percentage was in India. The MDD, MMF, and CIRF percentages were also low, 24%, 40%, and 23%, respectively. The scenario is similar for individual SACs as well. There are several reasons for the variations in complementary feeding (ISSSF, MDD, MMF, MAD, and CRIF) practices in SACs. These include maternal knowledge, conception, religious belief, cultural beliefs, differences in socio-economic and cultural characteristics [36,37,38]. An Indian study revealed that female elderly persons insisted mothers start complementary feeding only after one year for their children [39]. Another study in Pakistan revealed a lack of maternal knowledge and cultural beliefs as major barriers to complementary feeding [40].

We found higher breastfeeding practices among rural mothers (EBF, CBF), mothers from the poorest households, and mothers with secondary or higher education (EIBF, EBF). In contrast, lower complementary feeding practices were observed in the poorest and rural households. These findings are consistent with other studies from Bangladesh, India, Nepal, Sri Lanka, and Pakistan [12,41]. The low breastfeeding practices in urban and the richest households might be the employment of the urban mothers [42,43,44] and the affordability to buy BMS in the richest households [30,32]. Excessive workload, caesarean section delivery, discouragement and criticism inadequate creche facilities at the workplace, inadequate caregivers at home, inadequate knowledge about expressed milk are barriers to EBF for employed mothers [44,45]. Evidence suggested a baby-friendly workplace for employed mothers was effective for improving EBF practices [46,47]. Also, better knowledge, education, counselling, home support, six-months maternity leave, and flexibility at work are the enablers of exclusive breastfeeding for employed mothers in LMICs [44,48]. However, the low rate of complementary feeding in rural and the poorest households might be explained by the low purchasing capacity of diversified foods due to low income [49,50]. Interventions to improve complementary feeding should focus on these two vulnerable groups. In addition, we found lower complementary feeding among mothers with no formal education or below secondary education. This finding is consistent with a Pakistani study [51]. The low rate of complementary feeding among mothers with no formal education or below-secondary educated mothers could be explained by their lack of knowledge regarding the appropriate time to start and feed diversified complementary foods to their children [52,53]. Additionally, unemployment and lack of decision-making power might be another reason for the low rate of complementary feeding among the less educated mothers [39].

Among the SACs, Maldives had better IYCF practices. The Maldives developed national IYCF guidelines for promoting appropriate IYCF practices, developed the capacity of health care providers by training them for providing counselling to the caregivers on breastfeeding and complementary feeding [54]. The Maldives also initiated the Integrated National Nutrition strategic plan (INNSP). This plan includes interventions on exclusive breastfeeding, complementary feeding, growth monitoring, and micronutrient supplementation, [54] created advocacy groups and provided refresher training among doctors, nurses who working in the obstetrics and pediatrics department in health facilities to promote EBF [55]. The INNSP of Maldives identified the determinants of inadequate nutrient intake by children including breastfeeding and complementary feeding and initiated different intervention strategies to address them [56]. Other SACs, especially Afghanistan, Pakistan, and India should strengthen their IYCF policies by identifying the gaps in their national policies and programs to achieve optimal IYCF.

This study has several strengths. Similar sociodemographic factors have been considered for subgroups analyses for both country-specific and pooled data. We used the latest nationally representative data and unique methodology of DHS for all countries which reflects the up-to-date scenario of IYCF indicators and makes the comparison more robust. The comparison of IYCF indicators among countries helped us to identify vulnerable countries in terms of IYCF practices in SACs. Additionally, the subgroup analyses in different countries helped us to recognize the vulnerable groups who need immediate attention. Finally, the nationally representative large sample size was used which makes the evidence representative and generalizable. This study has a few limitations as well. We used age-specific sub-samples for calculating different IYCF indicators which reduced the estimated sample size. However, these sub-samples were large enough for pooled and country-specific analyses to make the estimates representative. The IYCF indicators were calculated based on the 24-h dietary recall-based subjective response from the mothers that might have caused recall bias. Since WHO constructed the IYCF indicators based on these recall responses [3], it could be considered the best possible approach for a nationally representative survey context.

## 5. Conclusions

This study concludes that the prevalence of IYCF practices varied substantially between and within countries, with IYCF practices were suboptimal in Afghanistan, Pakistan, and India. The breastfeeding-related indicators were suboptimal among the richest, urban, and mothers with secondary or higher-level education. The complimentary feeding-related indicators were suboptimal among the poorest, rural, and mothers with no formal or below secondary level education. Designing and scaling-up of indicator-specific tailored interventions targeting the identified population who are most at risk need to be implemented to enhance IYCF practices. In addition, a surveillance system should be placed to continuously monitor the progress and gaps in IYCF practices between and within countries. Future research should consider the identification of other factors causing suboptimal IYCF practices for specific marginalized populations to better inform interventions and policies for upgrading IYCF practices in SACs.

## Figures and Tables

**Figure 1 ijerph-19-04350-f001:**
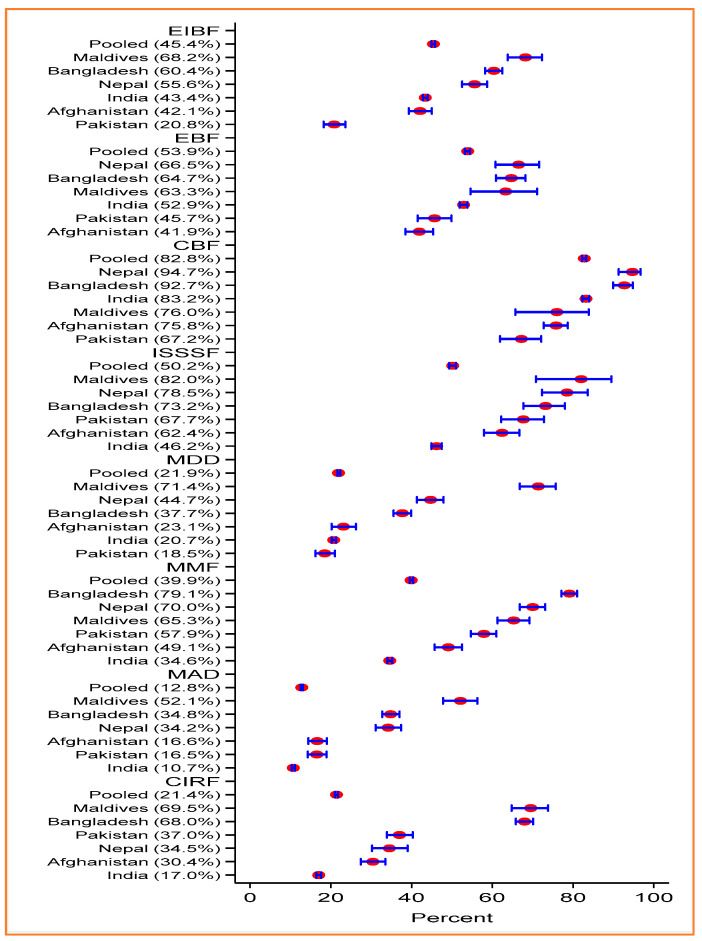
Prevalence of IYCF practices in South Asian countries.

**Figure 2 ijerph-19-04350-f002:**
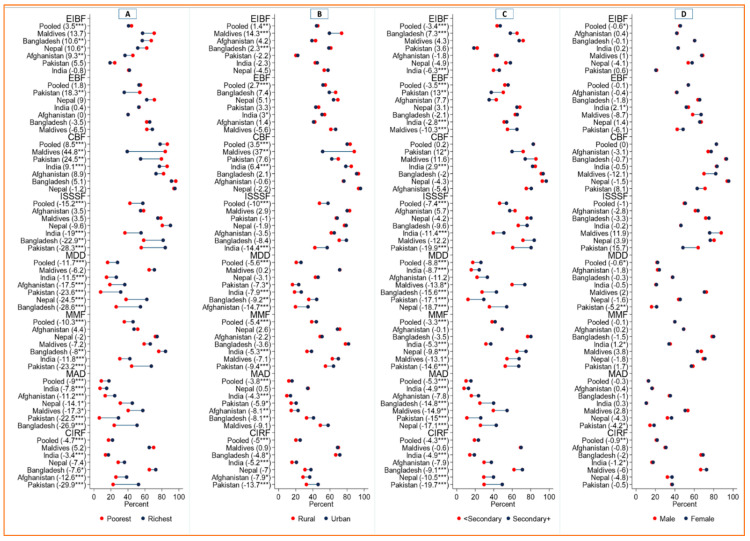
Absolute gaps in the prevalence of IYCF indicators across equity dimensions in South Asian countries. EIBF = Early initiation of breastfeeding, EBF = Exclusive breastfeeding, CBF = Continued breastfeeding at 1 year, ISSSF = Introduction of solid semi-solid and soft foods, MDD = Minimum dietary diversity, MMF=Minimum meal frequency, MAD = Minimum acceptable diet, CIRF = Consumption of iron-rich or iron-fortified foods. Note: Absolute gaps were calculated by subtracting the estimates/prevalence of IYCF indicators of the richest groups from the poorest (in panel (**A**)), of urban from rural (in panel (**B**)), of secondary+ education from below secondary education (in panel (**C**)) and of female from the male (in panel (**D**)). *** denotes *p*-value < 0.001, ** denotes *p*-value < 0.01, and * denotes *p*-value < 0.05.

**Figure 3 ijerph-19-04350-f003:**
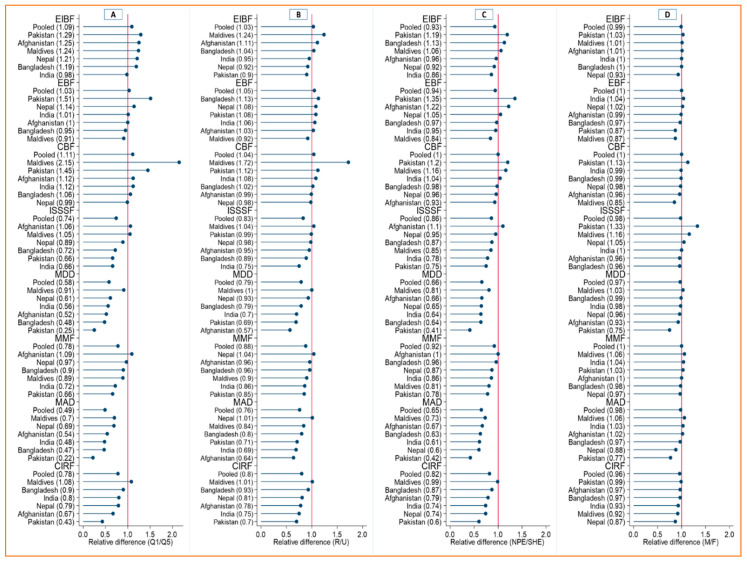
Relative gaps in the prevalence of IYCF indicators across equity dimensions in South Asian countries. EIBF = Early initiation of breastfeeding, EBF = Exclusive breastfeeding, CBF = Continued breastfeeding at 1 year, ISSSF = Introduction of solid semi-solid and soft foods, MDD = Minimum dietary diversity, MMF = Minimum meal frequency, MAD = Minimum acceptable diet, CIRF = Consumption of iron-rich or iron-fortified foods. Note: Relative gaps were calculated by dividing the estimates/prevalence of IYCF indicators of the poorest (Q1) groups by the richest (Q5) (in panel (**A**)), of rural (R) by urban (U) (in panel (**B**)), of below secondary education (NPE) by secondary+ education (SHE) (in panel (**C**)) and of male (M) by the female (F) (in panel (**D**)).

**Table 1 ijerph-19-04350-t001:** Socio-demographic characteristics of the study participants.

Characteristics	Pooled Data (2015–2018)	Afghanistan (2015 ^¥^)	Bangladesh (2017–2018 ^¥^)	India (2015–2016 ^¥^)	Maldives (2016–2017 ^¥^)	Nepal (2016 ^¥^)	Pakistan (2017–2018 ^¥^)	*p*-Value
	*n* (%) *n* = 120,830	*n* (%) *n* = 11,762	*n* (%) *n* = 3411	*n* (%) *n* = 97,935	*n* (%) *n* = 1147	*n* (%) *n* = 1919	*n* (%) *n* = 4656	
Mother’s Age								
15–24 years	52,217 (43.2)	4293 (37.6)	1883 (55.1)	43,310 (47.0)	258 (20.1)	1034 (52.6)	1439 (31.6)	<0.001
25–34 years	58,794 (48.7)	5454 (44.8)	1328 (39.3)	48,028 (47.5)	704 (61.9)	786 (42.2)	2494 (54.5)	
≥35 years	9819 (8.1)	2015 (17.7)	200 (5.6)	6597 (6.7)	185 (18.1)	99 (5.2)	723 (13.9)	
Mother’s Education								
<Secondary	57,216 (47.4)	10,751 (89.3)	1174 (34.0)	41,308 (40.8)	190 (16.4)	900 (48.7)	2893 (63.1)	<0.001
≥Secondary	63,614 (52.6)	1011 (10.7)	2237 (66.0)	56,627 (59.2)	957 (83.6)	1019 (51.3)	1763 (36.9)	
Child Sex								
Male	63,146 (52.3)	6111 (52.0)	1769 (52.0)	51,299 (52.5)	574 (59.6)	1038 (53.8)	2355 (50.3)	0.280
Female	57,684 (47.7)	5651 (48.0)	1642 (48.0)	46,636 (47.5)	573 (50.4)	881 (46.2)	2301 (49.7)	
Child age, mean (sd)	11.3 (6.6)	10.6 (6.6)	11.2 (6.9)	11.4 (6.6)	11.7 (6.8)	11.8 (6.8)	10.8 (6.6)	<0.001 *
Place of Residence								
Urban	30,159 (25.0)	2920 (24.2)	1167 (26.7)	22,790 (27.5)	114 (9.9)	1089 (53.6)	2079 (33.1)	<0.001
Rural	90,671 (75.0)	8842 (75.8)	2244 (73.3)	75,145 (72.6)	1033 (90.1)	830 (46.5)	2577 (67.0)	
Wealth Quantile								
Poorest	29,792 (24.7)	2019 (17.8)	724 (20.5)	25,225 (24.8)	311 (27.1)	486 (21.0)	1027 (21.5)	<0.001
Poorer	28,187 (23.3)	2667 (20.0)	712 (21)	23,044 (22.0)	327 (28.5)	407 (21.2)	1030 (19.1)	
Middle	24,750 (20.5)	2676 (21.5)	627 (19.2)	19,765 (20.2)	328 (28.6)	413 (23.0)	941 (21.5)	
Richer	20,959 (17.3)	2654 (21.5)	684 (20.5)	16,336 (18.1)	120 (10.5)	371 (20.5)	794 (18.6)	
Richest	17,142 (14.2)	1746 (19.3)	664 (18.8)	13,565 (14.8)	61 (5.3)	242 (14.4)	864 (19.3)	

^¥^ Survey year; sd = standard deviation; * *p*-value was generated from one-way analysis of variance; other *p*-values were generated from Chi-square test.

## Data Availability

The data of this study are publicly available. Data can be downloaded at https://dhsprogram.com/data/available-datasets.cfm (accessed on 3 April 2020).

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
