# Peer review of "Between and Within-Country Variations in Infant and Young Child Feeding Practices in South Asia"

_ijerph, 2022, doi:10.3390/ijerph19074350_

Round 1

Reviewer 1 Report

This is appropriate topic to this journal and provides and helpful insight about child feeding pratices in Africa, taking into account there is a dearth of evidence and research on all core IYCF indicators at regional or country levels, butsome revisions are needed to be clearer and to improve the quality of the article.

Overall the literature review, conceptual framework and purpose of article focus the main topic of the article, but it would be helpful for the authors to read through again with the intention of being even more precise and documenting with more recent and ongoing research on evidenced based related to feeding practices. Also if the study aimed to investigate the relationship between sociodemographic characteristics including place of residence, mother age, mother education, child sex, and wealth quintiles within the SAC , these should be explore/present in the background/introduction. I feel that is necessary to go deep in these characteristics

Methods: become clearer how was calculated different IYCF indicators. Because it looks like it reduces the estimated sample size; also explain how this not compromise the aim of the present study.

Became clearer the exclusion criteria

Results and discussions: Some findings are poorly support with literature (it is just made references to one author/study… this is not sufficient) and they should consider and discuss the variables that are included in the introduction.

Conclusion: I did feel a need for a deeper conclusion at the end of article that summarizes the principles themes/objectives that appeared in the introduction section - weaving back to recent highlights in the literature.

Recommendations/clinical implications: would be helpful to have specific recommendations for future research and practice taking into account this study and what was your aims,

Organization of the paper, grammar, and references: Another read through to check grammar errors and format would be helpful. Rewrite some parts to be more clear and use a more scientifc languague and appropriate english.

Author Response

Comments and Suggestions for Authors

This is appropriate topic to this journal and provides and helpful insight about child feeding pratices in Africa, taking into account there is a dearth of evidence and research on all core IYCF indicators at regional or country levels, butsome revisions are needed to be clearer and to improve the quality of the article.

Overall the literature review, conceptual framework and purpose of article focus the main topic of the article, but it would be helpful for the authors to read through again with the intention of being even more precise and documenting with more recent and ongoing research on evidenced based related to feeding practices. Also if the study aimed to investigate the relationship between sociodemographic characteristics including place of residence, mother age, mother education, child sex, and wealth quintiles within the SAC , these should be explore/present in the background/introduction. I feel that is necessary to go deep in these characteristics

Response: Thank you for your valuable comment. This study aimed to explore the between and within-country variations of IYCF practices and disparities of IYCF practices across different sociodemographic equity dimensions but not to investigate the relationship between socio-demographic variables with IYCF indicators. Therefore, we didn’t include the details about sociodemographic characteristics in the introduction section. However, according to your suggestion, we have mentioned the disparities of all sociodemographic variables included in our study in the introduction section in the revised version (page 2, line: 64-66).    

Methods: become clearer how was calculated different IYCF indicators. Because it looks like it reduces the estimated sample size; also explain how this not compromise the aim of the present study.

Response: We mentioned the detailed description and calculation of IYCF indicators in the supplementary Table 1. We completely agree with you that for calculating different IYCF indicators we needed the age-specific sample size which reduced the estimated sample size. We calculated these IYCF indicators based on the WHO calculation references. We believe that, though the sample size for some indicators such as exclusive breastfeeding, continued breastfeeding at one year and introduction of solid, semi-solid or soft foods decreased but still these samples have enough representativeness since the DHS uses a unique methodology and rigorous techniques for collecting the data.    

Became clearer the exclusion criteria

Response: According to your suggestions, we have included the exclusion criteria in the study population sub-section (page 1, line: 93-95).

Results and discussions: Some findings are poorly support with literature (it is just made references to one author/study… this is not sufficient) and they should consider and discuss the variables that are included in the introduction.

Response: According to your suggestions, we have added more recent literature in the discussion section which we also think will make the evidence stronger. We have already discussed our studied variables including mother’s education, wealth quantities, place of residence in the discussion section. We didn’t discuss the child’s sex and mother’s age variables because in most of the cases, we found insignificant differences with IYCF indicators for these two variables.    

Conclusion: I did feel a need for a deeper conclusion at the end of article that summarizes the principles themes/objectives that appeared in the introduction section - weaving back to recent highlights in the literature.

Response: We have revised the conclusion through providing a deeper and specific evidence-based conclusion. The revised conclusion now reads as following (page 10, lines: 320-331):

“This study concludes that the prevalence of IYCF practices varied substantially between and within countries, with IYCF practices were suboptimal in Afghanistan, Pakistan and India. The breastfeeding-related indicators were suboptimal among the richest, urban and mothers with secondary or higher-level education. The complimentary feeding-related indicators were suboptimal among the poorest, rural and mothers with no formal or below secondary level education. Designing and scaling-up of indicator-specific tailored interventions targeting the identified population who are most at risk need to be implemented to enhance IYCF practices. In addition, surveillance system should be placed to continuously monitor the progress and gaps in IYCF practices between and within countries. Future research should consider the identification of other factors causing suboptimal IYCF practices for specific marginalized populations to better inform interventions and policies for upgrading IYCF practices in SACs.

Recommendations/clinical implications: would be helpful to have specific recommendations for future research and practice taking into account this study and what was your aims,

Response: We have revised the conclusion and provided indicators for future research to better respond the interventions and policies. The added sentences are as following (page 10, lines 327-331):

“In addition, surveillance system should be placed to continuously monitor the progress and gaps in IYCF practices between and within countries. Future research should consider the identification of other factors causing suboptimal IYCF practices for specific marginalized populations to better inform interventions and policies for upgrading IYCF practices in SACs.”

Organization of the paper, grammar, and references: Another read through to check grammar errors and format would be helpful. Rewrite some parts to be more clear and use a more scientifc languague and appropriate english.

Response: Thank you so much for your valuable suggestion. We had carefully gone through the manuscript and tried to correct the grammatical errors, format, and re-written the conclusion section.

Reviewer 2 Report

Dear authors, thank you for this interesting and thorough manuscript. I have only a few comments that will require some consideration. Overall, I thought the paper was clearly expressed and the results well-presented and I think it will be of interest to readers. 

Discussion
I thought some comments in the discussion needed support by evidence, or at least an acknowledgement if there is no evidence to support it.
Lns 227-229 "Intensifying the market 227 monitoring system in each SACs with proper budget and trained manpower could be a 228 way to enforce the BMS laws and policies and could improve breastfeeding practices."

Lns 256-257 "However, the low rate of complementary feeding in rural and the poorest households 255 might be explained by the low purchasing capacity of diversified foods due to low income." 

Lns 265-274 Any evidence of the effectiveness of the Maldives National Nutrition strategic plan? It can be included here to support your point.

Formatting
Abbreviation IYCF needs to be in full words in the abstract
Reference list needs to be in the journals reference style

Author Response

Comments and Suggestions for Authors

Dear authors, thank you for this interesting and thorough manuscript. I have only a few comments that will require some consideration. Overall, I thought the paper was clearly expressed and the results well-presented and I think it will be of interest to readers. 

Response: Thank you for your time and review the manuscript.

Discussion
I thought some comments in the discussion needed support by evidence, or at least an acknowledgement if there is no evidence to support it.

Lns 227-229 "Intensifying the market 227 monitoring system in each SACs with proper budget and trained manpower could be a 228 way to enforce the BMS laws and policies and could improve breastfeeding practices."

Response: Cited a reference (ref. no 34) (page 8, line: 245).

Lns 256-257 "However, the low rate of complementary feeding in rural and the poorest households 255 might be explained by the low purchasing capacity of diversified foods due to low income." 

Response: We have included two references (ref. 49, 50) (page 8, line: 273).

Lns 265-274 Any evidence of the effectiveness of the Maldives National Nutrition strategic plan? It can be included here to support your point.

Response: According to your suggestion, we have included the effectiveness evidence of Maldives national nutrition strategic plan with a reference (ref. 56) (page 9, line: 292).

Formatting
Abbreviation IYCF needs to be in full words in the abstract
Reference list needs to be in the journals reference style

Response: Thanks. Spelled-out IYCF in the abstract. We have re-arranged the references according to the journal style in the revised version.